# Childhood Obesity Prevention in Africa: A Systematic Review of Intervention Effectiveness and Implementation

**DOI:** 10.3390/ijerph16071212

**Published:** 2019-04-04

**Authors:** Sonja Klingberg, Catherine E. Draper, Lisa K. Micklesfield, Sara E. Benjamin-Neelon, Esther M. F. van Sluijs

**Affiliations:** 1UKCRC Centre for Diet and Activity Research (CEDAR), MRC Epidemiology Unit, University of Cambridge, Cambridge CB2 0QQ, UK; ev234@medschl.cam.ac.uk; 2MRC/Wits Developmental Pathways for Health Research Unit (DPHRU), Faculty of Health Sciences, University of the Witwatersrand, Johannesburg 1862, South Africa; catherine.draper@wits.ac.za (C.E.D.); lisa.micklesfield@wits.ac.za (L.K.M.); 3Department of Health, Behavior and Society, Johns Hopkins University, Baltimore, MD 21205, USA; sara.neelon@jhu.edu

**Keywords:** low- and middle-income countries (LMIC), behavioural intervention, physical activity, dietary behaviour, sedentary, school setting, intervention evaluation

## Abstract

Childhood obesity is of increasing concern in many parts of Africa. We conducted a systematic search and review of published literature on behavioural childhood obesity prevention interventions. A literature search identified peer-reviewed literature from seven databases, and unindexed African journals, including experimental studies targeting children age 2–18 years in African countries, published in any language since 1990. All experimental designs were eligible; outcomes of interest were both behavioural (physical activity, dietary behaviours) and anthropometric (weight, body mass index, body composition). We also searched for process evaluations or other implementation observations. Methodological quality was assessed; evidence was synthesised narratively as a meta-analysis was not possible. Seventeen articles describing 14 interventions in three countries (South Africa, Tunisia and Uganda) were included. Effect scores indicated no overall effect on dietary behaviours, with some beneficial effects on physical activity and anthropometric outcomes. The quality of evidence was predominantly weak. We identified barriers and facilitators to successful interventions, and these were largely resource-related. Our systematic review highlights research gaps in targeting alternative settings to schools, and younger age groups, and a need for more rigorous designs for evaluating effectiveness. We also recommend process evaluations being used more widely.

## 1. Introduction

Childhood obesity is an urgent global public health concern, with implications both for the physical and emotional wellbeing of children, as well as risks for health later in life [1,2]. Specifically, overweight or obesity in childhood is likely to persist into later life, and to lead to health problems such as hypertension, heart disease, and type 2 diabetes [3,4,5,6,7]. It is also associated with adverse psychosocial effects [8], and lower educational attainment [9]. The evidence base on childhood overweight and obesity, and programmes for their prevention, builds predominantly on research from high-income settings. Although the within-country proportion of children with overweight and obesity is higher in high-income countries than low- and middle-income countries (LMICs), the vast majority (87%) of children under the age of five years with overweight or obesity live in LMICs [10]. In Africa, the prevalence of overweight and obesity among children under five years of age was 5% in 2017, and in absolute numbers there has been an increase of almost 50% since 2000, from 6.6 million to 9.7 million in 2017 [10]. According to global analyses, obesity among 5- to 19-year-olds increased in every region of the world between 1975 and 2016, but the proportional rise was smallest in high-income regions (averaging 30–50% per decade), and largest in southern Africa (about 400% per decade) [1]. Moreover, some parts of the African continent are more severely affected than others, as in 2017 the prevalence of overweight or obesity among children under five years of age in North Africa and Southern Africa was 10.3% and 13.7% respectively [10]. Nutrition and physical activity transitions are complex in many African settings given that overweight and obesity are joining rather than replacing the earlier problems of malnutrition, and more evidence of how transitions are occurring along with societal changes such as urbanisation is needed [11,12,13,14]. While undernutrition still constitutes a major challenge across the African continent, with the prevalence of stunting at 30.3% and wasting at 2.1% in 2017 [10], obesity prevention now also warrants attention in Africa [14,15].

Earlier reviews of relevance to childhood obesity prevention with a focus on evidence from LMICs (including African countries) have examined school-based childhood obesity prevention [16], physical activity promotion [17], and other obesity-related topics, such as the relationship between socioeconomic status and overweight and obesity, among school children in Sub-Saharan Africa [18]. Recent reviews of high-quality childhood obesity prevention interventions shed light on effective interventions from many different settings but do not include any studies from African countries [19,20,21]. While this may mean that there is a dearth of high-quality evidence from the African continent, it is critical to understand the research that has already been conducted in the region, and how this can inform future interventions. This can help identify and assess current research gaps, and provide actionable recommendations for future research on a topic of increasing policy relevance for African countries. The best avenues for intervention cannot be assumed based on evidence from other countries or settings, particularly considering the coexistence of other forms of malnutrition with childhood overweight and obesity in Africa [10].

Patterns of malnutrition and related behaviours are shaped by factors like socioeconomic status, ethnicity, age, and gender [3,22,23,24]. Specifically, in African settings, there are many contextual barriers to healthy behaviours, such as the high cost of healthy foods [25,26], gendered notions of acceptable behaviour, body image and beauty ideals [27,28,29,30], limited resources for sports and physical activity [31], de-prioritisation of physical education in schools [31,32], and safety concerns restricting physical activity [28,33,34]. It is therefore also relevant to consider whether interventions are targeting individual behaviours directly, or through some aspect of the environment. The social ecological model conceptualises different levels of the environment in relation to individuals, and is thus a useful framework for examining behavioural interventions, and how they are situated within the wider context [35]. These levels include individual or intrapersonal, interpersonal, institution, community and policy.

Systematic reviews have been criticised for inadequately considering the context in which interventions take place [36], and we thus we sought to maintain a reflective approach to context-specific findings throughout. Considering the diversity of countries classified as low- and middle-income, this review focuses on a specific geographical region, while also recognising that findings may still not be applicable across the region due to the complexities and contextual factors of different health systems and communities. 

The aim of this paper is to review existing evidence on the effectiveness of behavioural childhood obesity prevention interventions in African countries on anthropometric and behavioural outcomes in children ages 2–18 years. In addition, the following sub-questions were considered: What behaviours have been addressed in past interventions?What age groups and settings have the interventions targeted?What levels of the social ecological model are the interventions situated within?How do these aforementioned characteristics relate to effectiveness of interventions?What barriers and facilitators to implementation or effectiveness have been identified in existing studies?

## 2. Materials and Methods

We conducted and report our systematic review according to the PRISMA guidelines [37], and focused on behavioural childhood obesity prevention interventions targeting 2–18 year-olds in African countries. We only considered published, peer-reviewed articles describing experimental or quasi-experimental studies. We did not apply any limitations to language but considered literature published before 1990 unlikely to be of relevance due to the low levels of overweight and obesity prevalence in the African region at that time [38]. Table 1 describes all inclusion and exclusion criteria.

We employed a comprehensive search strategy using the search terms outlined in Appendix A to identify relevant literature published between January 1990 and May 2017. We searched the following seven databases: Embase, Scopus, Medline, Web of Science, SciELO, PsycINFO, and the Cochrane Library.

In addition to database searches, we searched for additional relevant literature through checking references of included articles, and the previously mentioned existing reviews [16,17,18], consulted key researchers regarding relevant journals to screen, and screened online archives of recommended regional journals (see Appendix A). This screening process covered all issues available in the journals’ online archives as of June 2017. 

We used referencing software (Mendeley, EndNote) to manage titles and abstracts retrieved through the comprehensive search. We removed duplicates, and screened citations using a checklist based on the eligibility criteria. One reviewer undertook all title and abstract screening, with another carrying out a duplicate screening of 500 titles in order to harmonise screening approaches. A third reviewer further checked a random sample of 10% of all titles and abstracts. We obtained the full text of all studies identified as potentially eligible following the screening of titles and abstracts. Two reviewers duplicate screened these articles independently, and decided on final inclusion based on the eligibility criteria in Table 1. The flow of the screening and selection process is illustrated in Figure 1.

One reviewer carried out data extraction using a piloted data extraction spreadsheet, and a second checked the extracted data adding any missing information to the form. We extracted the following data: study title, intervention name, population targeted, intervention description, study design, information about control group or comparison, outcomes, outcome measurements, publication type, publication year, setting, country, language, inclusion criteria, baseline descriptive data, randomisation procedure, length of intervention, length of follow-up, number of follow-ups, losses to follow-up, sample size, effectiveness for all relevant outcomes, details of tests and adjustment, subgroup effects if relevant, and any related publications referred to in the article.

Two reviewers independently carried out a duplicate scoring exercise, assigning effect scores to each outcome type (dietary behaviour, physical activity, or anthropometric outcomes) for each intervention. This approach to comparing effects has been used and described in other reviews [39,40,41,42]. The scores were composites based on all relevant outcomes (e.g., different fitness indicators) reported in each study, and ranged from “++” to “−−“, where “++” denotes a statistically significant, clearly intervention-attributable desired change on primary outcome or most outcomes of interest; “+” denotes a desired change on primary outcome, or mostly desired changes on relevant outcomes; “0” denotes no changes, mostly no changes, or both positive and negative changes that cancel each other out; “−“ denotes a negative change on primary outcome, or mostly negative changes on relevant outcomes; and “−−” denotes a statistically significant, clearly intervention-attributable negative change on primary outcome or most outcomes of interest. The two authors re-examined differences in scoring, ensuring that each intervention was scored according to the scoring criteria. They also generated summary scores for each behaviour by comparing the number of different scores and awarding the most frequent score as the summary score.

Two reviewers independently carried out a duplicate assessment of the methodological quality of included studies using the Quality Assessment Tool for Quantitative Studies developed by the Effective Public Health Practice Project [43]. Where it was difficult to fully harmonise ratings due to unclear criteria or study reporting, a combined quality rating (e.g., weak-moderate) was assigned. No meta-analysis was carried out due to heterogeneity in study designs, outcome measures, and reporting.

In order to answer the sub-questions of this review, one author conducted a search and review of process evaluations or other articles related to the included interventions (“sibling articles”). Search approaches included reference checking of included articles and using intervention names and author names as search terms in PubMed and Google Scholar. Sibling articles identified through this process, as well as all included articles, were then re-reviewed and data corresponding to the review’s sub-questions were extracted using an Excel spreadsheet. These data items included behaviours, age groups, and levels of the social ecological model targeted by each intervention, observations (including quotes) about how these characteristics relate to effectiveness, as well as observations regarding barriers and facilitators to intervention effectiveness or implementation. Data extraction and the resulting observations were checked by two other authors, and reflections were discussed within the review team.

## 3. Results

### 3.1. Design and Quality of Included Interventions

The combined search strategies yielded 9,714 non-duplicate articles, of which 17 were included for full review (Figure 1). They describe 14 different interventions from three African countries: South Africa (*n* = 9), Tunisia (*n* = 4), and Uganda (*n* = 1). Articles that were excluded during the full text screening stage were either targeting undernutrition, did not test a behavioural intervention, or were evaluated outside of Africa. Included interventions and their evaluations are described in detail in Table 2. Out of the 14 included interventions, three were randomised controlled trials, while the others utilised pre-test/post-test designs with (*n* = 8) or without (*n* = 3) a comparison group. The methodological quality of most studies was considered weak (*n* = 11), and this was due to both shortcomings in design and incomplete reporting. The theoretical basis of interventions was seldom explicitly reported but some (*n* = 3) referred to the social ecological model or social cognitive theory [46,47,48,49]. 

Very few formal process evaluations or reflective sibling articles were identified through the additional search process. Only one intervention (HealthKick, tested in the Western Cape in South Africa) explicitly involved a process evaluation, and there are several published articles documenting everything from intervention development [32,33,50] to implementation [51] of the HealthKick intervention. Moreover, the authors of some of the other included studies provided useful insights about the interventions or the study context more generally either in the evaluation studies included in the systematic review [52,53,54,55,56], or in other publications [31,57,58].

### 3.2. Targeted Settings, Age Groups and Behaviours

All except one preschool intervention in Tunisia and one community-based sport-for-development programme in Uganda were school-based or after school programmes. Five interventions reportedly targeted more than one setting: school and family [54,56], preschool and family [59], school and community [52], and school, family and community [60]. Included interventions addressed physical activity (*n* = 12) [47,48,49,53,54,55,56,59,60,61,62,63], dietary behaviour (*n* = 6) [46,48,49,59,60,64,65], and eight reported on anthropometric outcomes [52,53,54,55,60,62,63,66]. Only one intervention [59] targeted preschool-age children, while all others targeted school-age children. 

### 3.3. Outcome Measures

A diverse range of outcome measures were employed across the studies (Table 2 and Appendix A). Although many studies reported on physical activity, most relied on subjective, self-report data as opposed to objective measurements. Only one intervention evaluation used accelerometery data [56]. Evaluation of fitness was generally done using recognised protocols. Dietary behaviours were similarly generally measured using self-report, or parents’ reports of children’s diets. Included studies did not generally report on whether measures or protocols had been validated in the specific context in which they were being used. Shortcomings in reporting of outcome measures contributed to low quality ratings of many studies.

### 3.4. Intervention Characteristics and Levels of the Social Ecological Model

Interventions included curriculum changes [48,53,54,55,62], additional sessions of physical activity or physical education [46,47,48,49,53,54,59,60,61,63,64,66], additional teaching around healthy eating and lifestyles in general [46,47,48,49,52,53,54,59,60,64,65], providing training or materials to teachers or parents [46,47,53,54,55,56,59,60,62,64], organising sports tournaments or leagues [48,60,63], providing or improving school meals [52], and changing different aspects of the school environment [46,47,52,56,60,62,64]. All but one intervention involved several different components, the exception being a low-cost physical activity promotion intervention that primarily involved providing sports equipment, toys, and upgrades to the school playground in order to stimulate more free play [56]. As for levels of the social ecological model [35], most interventions focused on individuals, and to a lesser degree school environments, including school level policies and curricula (institutional level). Some also targeted teachers and families (interpersonal level) and the community. Intervention length varied from six days to three years. Intervention-specific levels of the social ecological model and estimated intervention doses are reported in Table 2.

### 3.5. Effectiveness

Table 3 summarises intervention effects by intervention (See Appendix A for further details on effects for specific outcomes). There was no overall evidence of effect on dietary behaviour. Only two studies [49,65] out of six reported an overall positive effect, while the remaining four interventions reported no effects [46,48,59,60,64]. More studies reported improvements in physical activity, and particularly fitness, with 6 of 12 interventions reporting positive overall effects [48,49,53,55,56,62]. However, the remaining studies reported no overall effects [47,54,59,60,63,66], resulting in an overall physical activity effect score of between “0” and “+”.

Beyond behavioural outcomes, positive effects on anthropometric outcomes (*n* = 4 of 8 studies) included reductions in the prevalence of overweight or obesity (−3.1 percentage points [60], −7.4 percentage points [52]), and mean weight [54], and a statistically significant, intervention-attributable reduction in body fat in one South African study [66]. However, one study reported an increase in weight among participants in the intervention group [53], which was statistically significant when compared to the control group. In summary, the overall effect score for anthropometric outcomes was between “0” and “+”.

In terms of targeting or reaching specific groups, a study evaluating a South African physical activity intervention included observations regarding the differential effects the intervention had in different age groups [56]. The strongest effect on physical activity outcomes was found in the youngest age group (Grade 3 learners, mean age 9.22) compared to the other groups (Grade 4–6 learners, mean age 10.42–11.45), and this was interpreted to be because the intervention promoted physical activity in the form of playing that may have been more suitable for the younger children. Moreover, the Ugandan sports-for-development programme seemed to attract participants who were already physically fit, and thus failed to target those who would have benefitted most from the intervention [57]. 

While not all school-based or after school interventions (*n* = 12) were successful or effective, their pooled effect scores are clearly positive for both physical activity and anthropometric outcomes when separated from the non-school-based interventions (*n* = 2), neither of which were effective. The effect score for dietary behaviours remains at zero when looking at intervention settings separately.

### 3.6. Implementation Barriers and Facilitators

Barriers reported to implementing school setting interventions included a lack of resources in schools [51,53,54,56], low priority of physical education [56], teachers’ or other stakeholders’ lack of time, buy-in, training or motivation [51,54,55], teachers’ fear of being criticised for implementing an intervention incorrectly [53], and external disruptions to implementation, such as strikes [51]. 

Two South African school-based studies recognised teachers’ positive attitudes towards the interventions as facilitating implementation. One study by Walter reports that a physical activity intervention which involved giving low resource schools playground and sports equipment to stimulate free play also improved the appearance of the schools, and teachers reportedly responded positively to the changes, with comments such as “our school looks like a real school now” [56] (p. 364). The qualitative evaluation of the South African Healthnutz intervention found that teachers were observing positive changes to the school environment, such as improved dynamics between teachers and learners, as a result of the intervention [53]. However, the authors also reported implementation challenges, such as problems with motivating teachers to deliver the intervention.

A South African school breakfast intervention, which improved learners’ anthropometric outcomes and reduced the prevalence of overweight and obesity, reported that serving learners breakfast seemed to promote physical activity in addition to introducing the healthy habit of eating breakfast, as teachers subjectively observed changes in learners’ levels of activity [52]. In this urban low resource setting, the opportunity to engage in one healthy behaviour seemed to enable other healthy behaviours, and this may have contributed to achieving desired changes in anthropometric outcomes. 

In the process evaluation of the South African intervention HealthKick, it was observed that engaging parents might be more effective in achieving changes, as mainly targeting individuals and the school environment did not achieve significant results [51]. This would involve addressing more levels of the social ecological model at once. Indeed, the formative research carried out for HealthKick did involve parents, and identified the need to target parents through interventions too [33]. However, the formative stage also indicated that parental engagement was low and challenging for schools in the study area. Another South African intervention, the Department of Health’s Health Promoting Schools initiative, included a family component, and was reportedly successful in engaging parents [54]. Parents’ active participation in group discussions improved over the course of the intervention, and the authors also report that parents’ own attitudes and behaviours around physical activity improved, which contributed to promoting physical activity among learners.

## 4. Discussion

This systematic review provides an overview of behavioural childhood obesity prevention interventions that have been implemented and evaluated in African countries. It also reflects on lessons that can be learnt from these studies in the absence of a mature and high-quality evidence base. Most of the African interventions we reviewed had some reported effect on either behavioural or anthropometric outcomes but overall, there was only limited evidence of effectiveness on each outcome of interest. Evidence of effectiveness is particularly scant for dietary behaviours. We observed more promising effects on physical activity and anthropometric outcomes but the majority of interventions we assessed were of weak methodological quality. In light of the overall weak quality of the studies, it is clear that there is a dearth of high-quality evidence of effective strategies to prevent childhood obesity in African settings. However, some interventions did achieve desired changes to health-related behaviours, which is meaningful regardless of effects on obesity per se. Focusing on promoting healthy behaviours may be more appropriate than specifically framing interventions as obesity prevention, particularly while undernutrition also persists in many African settings.

The included studies described interventions in only three different African countries, which in itself suggests that evidence of childhood obesity prevention interventions in Africa is scant. This partly reflects the comparably low rates of childhood overweight or obesity in some parts of Africa but it is interesting to compare the context of these three countries. Tunisia is a lower middle-income country in North Africa, with a reported overweight or obesity prevalence of 25% among 5–19-year-olds in 2016 [67]. In South Africa, an upper middle-income country, the corresponding figure was 24.7% in 2016 [68], whereas Uganda, a low-income country in East Africa, had a lower overall prevalence of overweight or obesity of 10.3% among 5–19-year-olds in the same year [69]. The burden of non-communicable diseases is high in all three countries, with dietary risks, high blood pressure, high fasting glucose, and malnutrition among the top ten causes of disability-adjusted life years [70,71,72] even if the urgency of childhood obesity per se is not the same across the countries. 

School-based and after school interventions dominate much of the literature on childhood obesity interventions globally [19,20,21], and indeed, out of the interventions we identified those set around schools were effective in many cases. However, this conclusion should be drawn cautiously considering that so few other intervention settings have been comprehensively tested in African countries. Based on our review, it is not possible to conclude that non-school-based interventions are ineffective even though these approaches did not demonstrate intervention-attributable effects in our synthesis. Further high-quality research is needed to identify the utility of targeting different settings in African contexts. 

As for which behaviours to target, there have been mixed results from other systematic reviews conducted in high-income settings. Interventions focusing on dietary behaviours alone were found to be more effective in a systematic review of childhood obesity prevention interventions set in high-income countries [19]. However, a global review which does not include any African countries found targeting dietary behaviours and physical activity together to be the most effective approach [20]. A synthesis of meta-analyses and reviews of health behaviour interventions suggests that interventions targeting single behaviours are more effective than those targeting multiple behaviours in changing the behaviour in question but interventions targeting multiple behaviours, such as both dietary behaviours and physical activity together may have a greater effect on changing weight [73]. Based on our review, there is no clear trend in existing African interventions to suggest that targeting single or multiple behaviours is better. Decisions about what behaviours to target should be guided by the behavioural epidemiology of the specific intervention context [74], and this may vary considerably between different age groups. More research on childhood obesity prevention interventions targeting younger, preschool-aged children in African settings is needed, given the rapidly increasing prevalence of overweight and obesity in this age group.

Moreover, other systematic reviews of childhood obesity prevention interventions suggest that focusing on schools in combination with community or home settings may be the best approach [19,20]. In our review, we found five examples of interventions targeting multiple settings [52,54,56,59,60], of which two targeted both schools and families [54,56], and one targeted schools, families and the community [60]. However, the observations regarding the challenges of engaging parents in school-based interventions [33,51] introduce the question of whether these combined approaches actually could improve the effectiveness of childhood obesity prevention interventions in low resource settings in African countries seeing as parental engagement may not be feasible. It is possible that a different approach is needed, and including a focus on community or wider family may be more useful than specifically trying to engage parents in school-based interventions. Again, understanding the specific intervention context is crucial when following recommendations from high-income country studies.

Nevertheless, some of the identified barriers (e.g., lack of buy-in or time from teachers or school management) and facilitators (e.g., intervention benefits beyond health outcomes) to implementation and effectiveness are similar to the challenges of implementing school-based interventions in high-income countries [53]. This may facilitate the use of existing, high-income country-dominated evidence of childhood obesity prevention interventions in designing and developing new interventions in LMIC settings. However, as most of the interventions included in our review are school-based, it is possible that interventions targeting other settings may encounter different challenges that are more unique to the specific context in which they are implemented. 

By assigning effect scores to each intervention by the behaviours it targeted, we were able to gain an overall understanding of the effectiveness of childhood obesity prevention interventions in African countries on anthropometric and behavioural outcomes in children ages 2–18 years. However, it was challenging to pool together effect scores for vastly different outcomes, such as specific fitness indicators and overall levels of physical activity. In most cases, it was also difficult to attribute effects to the interventions due to limitations in both study design, and analyses performed. Where detailed comparison between intervention and control groups was not available, we based judgements on what authors described in words, and cautiously assigned effect scores based on the data available in each paper. 

While the included studies were too heterogeneous for a meta-analysis, we found it useful to consider what evidence currently exists but has not necessarily been captured in global or regional reviews due to more stringent inclusion criteria. Another strength of this review is that the systematic search was complemented with a screening process of regionally relevant journals, which were not all indexed in mainstream databases. However, this also introduced an element of bias as searching and screening beyond databases involves choices around inclusion and exclusion of specific journals. Similarly, while we did not use language as an exclusion criterion, our search strategy was in English, and this may have affected our ability to find all potentially relevant articles in other languages if they did not have translated abstracts or English keywords.

The challenges we experienced in synthesising evidence due to the heterogeneity of studies have also been reported in other reviews focusing on African or LMIC settings [14,16]. By highlighting research gaps that exist both in terms of quality and quantity of studies, we hope to encourage further research into childhood obesity prevention interventions in African countries, including accompanying process evaluations. Conducting process evaluations, and using appropriate theories as the basis for interventions has been recommended but is often neglected [16,75,76]. While it cannot be claimed that theory-based interventions would have been more effective by virtue of being theory-based, a more explicit theoretical basis could have helped to unpack the limited effectiveness of many of the interventions included in this review, even in the absence of process evaluations. A key recommendation for future research is to make use of more rigorous evaluation designs in terms of appropriately powered randomised or cluster randomised controlled trials, as this would vastly improve the evidence base of childhood obesity prevention in African countries. However, this can only happen with substantial and sustained research funding, and further capacity building to ensure the availability of a well-trained work force to conduct the research. This may not only aid an increase in the quantity and quality of the research, but also the ability to develop and comprehensively evaluate more complex interventions, including those targeting families and communities. In addition to improving the quality of evaluative designs, and conducting process evaluations according to existing frameworks and guidance [75,76,77], more detailed reporting when publishing findings would also contribute to a higher quality of evidence.

## 5. Conclusions

Based on current evidence, school-based interventions have demonstrated some potential for childhood obesity prevention in South Africa and Tunisia, but our findings indicate limited overall effectiveness. However, there is a general lack of high-quality evidence of effective childhood obesity interventions across Africa, and we would be cautious about extrapolating from these findings when it comes to other African countries and intervention settings. In particular, few studies have targeted the family or community and even fewer have focused on very young children. Further research building on both global and context-specific evidence will help to develop more effective approaches to address childhood obesity as a growing public health concern.

## Figures and Tables

**Figure 1 ijerph-16-01212-f001:**
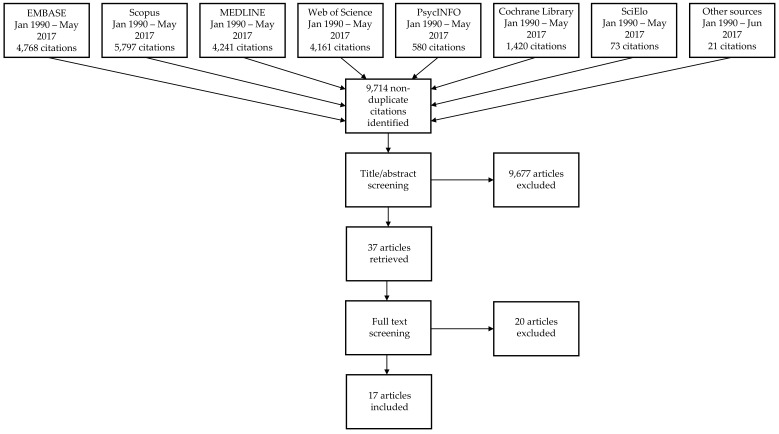
Flowchart of search, screening and selection processes.

**Table 1 ijerph-16-01212-t001:** Eligibility criteria for the systematic review.

	Included Studies	Excluded Studies
**Population**	Generally healthy, typically developing children and adolescents ages 2–18 years residing in African countries	Studies targeting children and adolescents with specific disease or condition, including asthma, diabetes, and obesity
	Normal or mixed weight populations	African populations residing outside of Africa
**Intervention**	Any behavioural (including but not limited to) physical activity- or diet-related interventions aimed at preventing overweight and obesity (even if not explicitly stated) among children in any context (home, community, school, etc.)	Obesity treatment interventions, malnutrition prevention interventions targeting undernutrition, non-behavioural interventions
**Study design**	Primary research question: Randomised or non-randomised controlled trials (cluster or individual), controlled pre-post studies, prospective cohort studies with a control group, interrupted time series and repeated measure studies, quasi-experimental studies and natural experiments	Cross-sectional studies, non-experimental studies, non-human studies, laboratory-based studies
	Sub-questions: Any design, including qualitative studies, as long as they are describing the same studies as those selected for answering the primary research question of the review	N/A
**Outcomes**	Primary outcomes: Adiposity-related outcomes, including prevalence of overweight and obesity, and body composition. Intermediate behavioural outcomes such as changes in physical activity and fitness, sedentary behaviour, and dietary behaviour	Other health outcomes, such as blood pressure, if not reporting about relevant adiposity outcomes
	For behavioural outcomes, both objective and subjective measures of physical activity, dietary behaviour, or other relevant behaviours, such as sedentary behaviour, are acceptable	Other outcomes of behavioural interventions, such as cognitive development, if not reporting about relevant behavioural outcomes (increased physical activity, fitness, sedentary behaviour, or dietary behaviour)
	Secondary outcomes from sibling article search: Barriers and facilitators to implementation of childhood obesity prevention interventions	N/A
**Publication type**	Peer-reviewed journal articles	Conference abstract, working paper, study protocol, report, dissertation, book, website
**Publication year**	1990 onward	Before 1990
**Setting**	Any African country according to the World Bank’s regional definitions of Sub-Saharan Africa and North Africa [44,45]	Countries in any other regions
**Language**	Any language	N/A

**Table 2 ijerph-16-01212-t002:** Characteristics of included childhood obesity prevention interventions in African countries (*n* = 14).

Intervention and Study References	Intervention Context (Targeted Setting)	Baseline Characteristics	Study Design	Components, Dose and Levels of Social Ecological Model	Outcomes
DoH Health Promoting Schools Nyawose & Naidoo 2016 [54]	Low socio-economic status Clermont Township, KwaZulu-Natal South Africa (School, family)	N = 129 Gender: 51.2% boys Age: 11–15, mean 12.26 years	Quasi-experimental, non-equivalent groups design with an intervention programme and assessment pre- and post- intervention.	4-month intervention. Introduced various methods of PA and healthy nutritional habits within the PE lessons in the school curriculum. A minimum of two one-hour PE workshops were conducted per month. Activities included warm-up games, circuit and fun group games. Parents took part in four group sessions where PA was discussed, and dietary guidelines were introduced. Unable to estimate overall dose received. Levels: Individual, interpersonal, institution.	Sports and PA participation (learner questionnaires that have been used in other South African studies), fitness (Eurofit Physical Fitness Test Battery adapted for use in South Africa), height and weight.
Gum Marom Kids League (GMKL) Richards et al., 2014 [63]	Post-conflict, urban low resource setting, Gulu, Uganda (Community)	N = 1462 Gender: 43.3% boys Age: 11–14	Single-blinded randomised controlled trial nested within observational study.	11-week voluntary competitive sport-for-development football league. 32 volunteer adults from the local community trained as football coaches. Each weekend the GMKL participants took part in a 40-min game of football and various peace-building activities. Overall dose: ~7.5 h over 11 weeks. Levels: Individual, interpersonal, community.	Physical fitness (multi-stage fitness test and standing broad jump), anthropometric outcomes (BMI-for-age and height-for-age z-scores compared with WHO reference data).
Harrabi et al., 2010 [48]	Secondary public schools in Sousse, Tunisia (School)	N = 2338 Gender: 46.8% boys Age: 12–16 (mean 13.3 ± 1.1)	Pre-test post-test quasi experimental design (with control group).	Intervention over one school year. Components included classroom-based health promotion, student projects, health clubs and discussions. Interventions were delivered by project team with teachers and school doctors. Interclass sport tournaments organised throughout the school year. Award ceremony held at the end. Unable to estimate overall dose received. Levels: Individual, interpersonal, institution.	Dietary habits and PA (pre-tested self-administered questionnaire).
HealthKick Steyn et al., 2015 [46], De Villiers et al., 2016 [64], Uys et al., 2016 [47]	Urban and rural primary schools from the lowest 3 socio-economic quintiles, Western Cape, South Africa (School)	N = 998 or 1002 Gender: 47.2% boys Age: 10 years at baseline	Cluster RCT.	3-year whole-of-school program targeting healthy eating and physical activity by creating a healthier school environment. Educators given training and resources to implement their own action plans. Educators asked to give extra 15 min of PA a day and at least one healthy eating activity per month. Schools set goals and implemented changes over three years. Dose: ~1.5 h/week for 3 school years. Levels: Individual, interpersonal, institution.	Dietary behavior (unquantified 24-h recall) and fitness (modified Eurofit). Used both validated and unvalidated questionnaires.
Healthnutz Draper et al., 2010 [53]	Poor urban school setting in Alexandra township, Johannesburg, South Africa (School)	N = Unclear Gender: NR Age: NR	Pre-post test (with control group).	3-month intervention. Training for teachers 2 months prior to implementation, weekly PA and health education sessions for learners incorporated into curriculum. Unable to estimate overall dose received. Levels: Individual, interpersonal, institution.	Anthropometric measurements (height and weight), physical fitness (Eurofit Fitness Testing protocol adapted for use in South Africa).
Hochfeld et al., 2016 [52]	Poor urban school setting in Alexandra township, Johannesburg, South Africa (School, community)	N = 1975 Gender: 52% girls Age: 6–17, median 10	Pre- and post-test design (no control group).	14-month intervention. School breakfast provided, school kitchen upgrades, nutrition education, community development activities. Unable to estimate overall dose received. Levels: Individual, institution, community.	Anthropometric measurements (height, weight, BMI using standard protocols).
Kebaili et al., 2014 [65]	Public schools in urban setting in Sousse, Tunisia (School)	N = 2338 Gender: I: 46.8% boys, C: 46.5% boys Age: 12–16	Pre-post quasi-experimental evaluation.	3-month intervention. Interactive lessons and activities delivered by trained teachers in collaboration with doctors. Unable to estimate overall dose received. Levels: Individual, interpersonal.	Dietary behaviour (pre-tested self-administered questionnaire).
Maatoug et al., 2015 [59]	Urban preschools in Sousse, Tunisia (Preschool, family)	N = 539 Gender: I: 53.6% boys, C: 46.4% boys Age: I: Mean 4.50 years (±0.51), C: 4.73 years (±0.34)	Quasi- experiment (with control group).	8-month preschool-based intervention. Lifestyle intervention with training sessions, workshops, tournaments and educative supports to teachers and parents. Unable to estimate overall dose received. Levels: Individual, interpersonal.	Eating habits, PA, and screen time (parent questionnaire).
“Masikhusele iKamva Lethu” (“Let Us Protect Our Future.”) Jemmott et al. [49]	Urban and rural schools in Eastern Cape, South Africa (School)	N = 1057 Gender: 52.8% girls Age: 9–18 (mean 12.4)	Cluster RCT.	6-day intervention. Theory-based, highly structured health promotion intervention consisting of 12 1-h modules. Sessions included interactive exercises, games, brainstorming, role-playing, and group discussions. Materials included comic workbooks specially designed for the intervention. Dose: 12 h in 1 week. Levels: Individual, interpersonal.	Dietary behaviour (self-report using 7-item food frequency questionnaire developed by the National Cancer Institute) and PA (self-reported PA over past 7 days using CDC-developed 3 item questionnaire).
Nutrition and Physical Activity (NAP) Pilot Naidoo et al., 2009 [62]	4 primary schools in KwaZulu-Natal, South Africa (School)	N = 256 Gender: 44% boys Age: Grade 6 learners	Prospective empirical pilot study with an intervention and an assessment before and after intervention (no control group).	6-month intervention. Classroom-based materials were developed with cost-effectiveness and sustainability in mind. NAP was integrated into the school curriculum. Educators were trained to lead intervention activities and had some freedom in how to implement these. At least two monthly follow-up visits to schools by the research team was provided. There were also changes to the school food environment. Unable to estimate overall dose received. Levels: Individual, interpersonal, institution.	PA (self-reported through learner questionnaire).
Nutrition and Physical Activity (NAP) Naidoo & Coopoo 2012 [55]	Rural, peri-urban and urban schools in KwaZulu-Natal, South Africa (School)	N = 798 at baseline Gender: 54% boys Age: 9–16 years (41% of learners age 12 at the onset of the study)	Pre-post evaluation (with control group).	18-month intervention. Classroom-based materials were developed with cost-effectiveness and sustainability in mind. NAP was integrated into the school curriculum. Educators were trained to lead intervention activities and had some freedom in how to implement these. Unable to estimate overall dose received. Levels: Individual, interpersonal, institution.	PA (self-reported through learner questionnaire) and fitness (measured using Eurofit Physical Fitness Test Battery, 1993).
PLAY Naude et al., 2008 [66] PLAY Lennox & Pienaar 2013 [61]	Secondary schools in a low socio-economic township area in the North-West Province, South Africa (After-school) Secondary schools in a low socio-economic township area in the North-West Province, South Africa (After-school)	N = 279 Gender: 40.5% boys Age: 13–18	Pre-post evaluation (with reference group).	19-week voluntary after school PA programme supervised by Biokinetics students. The programme was performed twice weekly for an hour session per day, and consisted of 20 min of aerobic dancing, 20 min of ball games, and 20 min of strength- and flexibility exercises. Dose: 38 h (2 h/week for 19 weeks) Levels: Individual.	BMI (anthropometric measurements according to ISAK-standard) and body fat % (Bod Pod, and tricep and subscapular skinfolds).
		N = 318 Gender: 43% boys Age: Grade 8 (13–14)	Quasi-experimental before-after evaluation (with control group).	6-month voluntary after-school physical activity intervention. Two 60-min sessions a week. The sessions were divided into 30 min of aerobic training, 15 min of strength and flexibility training, and 15 min of sport-related ball skills activities. Dose: 52 h (2 h/week for 26 weeks). Levels: Individual.	PA (previous day PA recall) and fitness (“The Bleep test”).
“Schools in Health” Maatoug et al., 2015 [60]	Urban school setting in Sousse, Tunisia (School, family, community)	N = 4003 Gender: I: 50.2% boys, C: 46.5% boys Age: 11–16	Quasi-experiment (with control group).	3-year school-based intervention. Trained student leaders organised events, teachers ran sessions to promote PA and healthy diets. After-school soccer games both within and between schools. Information about healthy behaviours was provided to students and parents. Snack stores were encouraged to stock healthier options, and children were rewarded with stickers for choosing healthy snacks. Unable to estimate overall dose received. Levels: Individual, interpersonal, institution, community.	Overweight/obesity (standard anthropometric measurements), PA (standardised, pretested questionnaire) and dietary behavior (standardised, pretested questionnaire).
Walter 2014 [56]	3 disadvantaged primary schools in Port Elizabeth, South Africa (School, family)	N = 79 Gender: 48.1% boys Age: Mean age 10.27 ± 1.22, range 9–12	Experimental design (no comparison).	6-week intervention delivered by University students with parents and teachers. The intervention focused around providing sports and play equipment to schools. Focus on free play. Unable to estimate overall dose received. Levels: Interpersonal, institution.	PA (Actigraph accelerometry).

BMI = Body Mass Index; C = control group; I = intervention group; NR = not reported; PA = physical activity; PE = physical education.

**Table 3 ijerph-16-01212-t003:** Quality assessment and effect scores by intervention and targeted outcome.

Study	Quality Assessment	Effect on Dietary Behaviours	Effect on Physical Activity	Effect on Anthropometric Outcomes
DoH Health Promoting Schools [54]	Weak	.	0	+
Gum Marom Kids League [63]	Moderate–strong	.	0	0
Harrabi et al. [48]	Weak	0	++	.
HealthKick [46,47,64]	Weak	0	0	.
Healthnutz [53]	Weak	.	+	-
Hochfeld et al. [52]	Weak	.	.	+
Kebaili et al. [65]	Weak–moderate	+	.	.
Maatoug et al. [59]	Weak	0	0	.
“Masikhusele iKamva Lethu” [49]	Weak	++	++	.
NAP pilot [62]	Weak	.	+	0
NAP [55]	Weak	.	+	0
PLAY [61,66]	Weak	.	0	++
“Schools in Health” [60]	Weak	0	0	+
Walter [56]	Weak–moderate	.	+	.
**Overall**	**Weak**	**0**	**0/+**	**0/+**

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
