# Peer review of "Childhood Obesity Prevention in Africa: A Systematic Review of Intervention Effectiveness and Implementation"

_ijerph, 2019, doi:10.3390/ijerph16071212_

Reviewer 1 Report

IJERPH-2-19

The major stated purpose of this manuscript is to provide a systematic review of  childhood obesity  (behavioural) prevention  interventions targeting children 2-18 years of age  in African countries.  Specified outcomes are anthropometrics and behaviours.  Questions that guided the review focus on the behaviours that have been addressed; the age groups and settings the interventions targeted; the levels of the socio-ecological model that were the settings for the interventions and barriers and facilitators to implementation or effectiveness .

 As presented, the manuscripts has several strengths  including the potential significance of the  review ( implications for future research in this area of inquiry). The use of the PRISMA framework  , the information presented in tables, the inclusion of some process evaluation measures, the evaluation of the quality of the studies and integration of relevant literature in the introduction and discussion sections are viewed as strengths. In addition, the inclusion of recent relevant data-based  and seminal conceptual references is/are also viewed as strengths.

This reviewer is delighted to note that challenges experienced in synthesizing evidence  ( due to the heterogeneity of studies ) is acknowledged as this is a major issue in summarizing intervention  research focused on obesity prevention and management in school based settings .

The authors are encouraged to add some specific directions for future research in this area of inquiry and relevant to LMIC countries-    In so doing, please consider the findings of this rigorous review as well as  the limitations and how future school-based efforts might optimize prevention and management of obesity in schools, an important population-based venue for risk reduction and health promotion with children, adolescents- and families.

Author Response

Thank you for these helpful comments. Our detailed response is in the attached cover letter.

Reviewer 2 Report

This is a very well written, timely and enlightening manuscript.
The only edits recommended are as follows:
1) The references listed for comorbidities (ie, ~ references 3-7, or so) are limited. There is a significant body of work that details associations and, in some instances, comorbidities with several conditions outside of type 2 diabetes (e.g., hypertension, obstructive sleep apnea, asthma (based on inflammatory type), depression and cancer (a decidedly long-term, adult-onset comorbid risk)). These should be referenced accordingly in a systematic review of this stature.
2) Lastly, in a systematic review of interventions it would be very helpful to have some description of the total contact time either reported directly by the individual studies, or as estimated by the authors' calculations (i.e., should align with contact estimates such as >26 hours in a 6 month period, as suggested by the United States Preventive ServicesTask Force). It would also be helpful to have some sense of the degree to which individual programs offered maintenance or booster doses of the intervention after completion of the primary intervention. Both of these additional details (i.e., contact time and nature of maintenance dose) could be included in a tabular format.

Author Response

(The authors gave the same response as above.)

Reviewer 3 Report

This review assessed an important and timely topic. I have several suggestions for improving the manuscript.

Introduction:

Lines 38-40: it would be helpful to know the context of this reference; for instance, since LMICs likely have a higher number of children overall, what are the proportional levels of obesity in low and high income countries? Is this adjusted for stunting? In the context of undernutrition and stunting, how should an obesity prevention intervention be different from a context where undernutrition is not a concern?

Lines 49-60: how does this review build upon and the previous systematic reviews on obesity and physical activity by Muthuri et al?

Lines 45-50 and 61-67: speaking more to the nutrition transition, the changing nutrition and physical activity landscape, and how these factors interact in the landscape would be helpful

Methods and results:

While a narrative review may have better aligned with the topic given the limited data, the systematic approach is strong. However, what is not addressed is the landscape of obesity in the three countries included in this study. Presumably, the prevalence of obesity, contributing factors, and environmental and cultural context would likely dictate the types of interventions that would be attempted. However, this context is not provided. Further, since the review covers only a limited number of studies, more context should be provided about the outcomes.

Rather than focusing on the quality of evidence (is this fair when there are so few studies, covering only three countries with very different contexts?), it would be helpful to focus more on the intervention design and outcomes by providing more context for the tables within the manuscripts.

Table 3 should include the study results written out to give an overview of the intervention results and allow readers to judge efficacy themselves.

Lines 256-286: More context should either be provided here or in the introduction on why these barriers and facilitators were relevant in the context of each study. In the discussion, there should be follow-up to how these differ from commonly-noted barriers/facilitators in other LMIC contexts and non-LMICs.

Discussion:

Line 297-299: should this perhaps be balanced with an understanding of possible negative consequences of obesity prevention interventions in a context where undernutrition still exists?  Why should the focus be on obesity rather than negative behaviors that potentially lead to obesity, particularly since obesity has such a low prevalence?

Lines 300-307: here and below, it would be helpful to go over the country-specific results, and explain again the prevalence of overweight and obesity in Tunisia, South Africa, and Uganda. What the proportion within each study?

Generally, could you speak more to the gaps, and how the interventions identified in this review differed from those in other LMICs? It isn’t clear from the discussion what type of intervention one should think about attempting, other than school-based might be helpful. Clear identification of gaps and the strongest evidence available would be helpful.

Author Response

(The authors gave the same response as above.)

Reviewer 4 Report

This review has been conducted in a proper way and the findings are relevant and interesting.I suggest that you add some discussion about the community centered cultures in Africa vs. individual centered in "high income countries". This difference may be takin into the consideration when planning an intervention for children? I also suggest to discuss more about the the age frame: do 2-year-olds need different intervention than e.g. teenagers?

Author Response

(The authors gave the same response as above.)

Reviewer 5 Report

Overall comments:

Thank you for the opportunity to review this manuscript. It addresses a gap in the literature that is of emerging public health concern in African countries, and the authors do a useful job of highlighting that need in the Introduction section of the paper.

Specific comments:

Abstract

·         The Abstract could be a little more refined. The purpose statement has elements of methods (eg eligibility criterion), and the results and methods are also mixed in this section (eg barriers and facilitators). Consider restructuring this section so the sections are clearly presented.

1. Introduction

·         Page 1, line 35: include “…persist into later life,…”

·         Page 1, line 37: reword the sentence to “…The evidence base on childhood overweight and obesity and prevention programs…”

·         Page 1, line 38: add “of” after “…(87%)…”

·         Page 2, line 51: state the specific reviews you are referring to ie “Earlier reviews…” of what? eg obesity prevention programs

·         The flow between the paragraphs on page 2 could be better (ie paragraph starting on line 61). Consider reworking these sections slightly to create a clearer more cohesive argument. The points being made are useful. The flow may need a little work.

·         Consider adding “…behavioural childhood obesity prevention interventions…”  to the aim of the paper from line 79 (page 2)

·         With the sub-questions, consider:

o   Changing point 2 to “sub-groups” rather than “age groups” given you also consider the existing levels of physical activity of participants in critiquing intervention effectiveness

o   Dot point 4, what does “these characteristics” refer to? Can this be clarified?

2. Materials and Methods

·         Page 3-4, Table 1:

o   “Intervention”: What is meant by “…behavioural determinants of overweight and obesity…”? Can this be defined?

o   “Study design”: prospective cohort studies by definition do not include an intervention. They are observational studies only. Given your review involves experimental/intervention studies, including cohort studies would be inappropriate. Did you use any cohort studies? If not, consider removing it from your inclusion criteria.

o   “Outcomes”: there is a mix of outcomes and measures presented in this section – eg prevalence and z-scores for some outcomes while the measures of other outcomes are not stated (eg BMI). z-scores are also usually reported with a mean. Noting the outcomes only in this section rather than how they are measured may be all that is required.

§  The first time you report BMI, state the abbreviation in full ie “body mass index (BMI)”

§  “Secondary outcomes”: are these related to the search for process evaluations of interventions specifically in the search strategy? It is a little unclear how this relates to the search strategy (as opposed to the information extracted from studies). Could this be clarified for the reader? This may also need clarifying on page 2, lines 92-95 where it states you only searched for experimental studies.

3. Results

·         3.2. Targeted settings, age groups and behaviours:

o   page 5, line 173; replace the “,” with a “.” after reference 46.

o   When reporting the number of interventions targeting diet and physical activity, and reporting anthropometric outcomes, include references of studies included in each.

·         Figure 1: suggest moving the description of how criteria were applied to the corresponding box; ie move “title/abstract screen” to “inclusion/exclusion criteria applied to title/abstract” and “full text” to “inclusion/exclusion criteria applied to full text”.

o   3.3 Outcome measures: page 6, line 178: change “was” to “were”.

·         3.4. Intervention characteristics and levels of the social ecological model

o   Page 6, line 189-190: suggest separating reporting of training of teachers separately to parents, given they are possibly different types of strategies with different potential effects.

o   Page 6, line 192: It states that “Most interventions involved several different components…” Is it all but one study? If so, suggest stating that and referencing those you are referring to.

o   When discussing the social ecological model, include references of which studies were included in each level.

·         3.5 Effectiveness

o   Page 6, line 201: consider stating up front “Only two studies out of six…reported an overall…”

o   Page 6, line 202-203: State “More studies reported improvements in physical activity…” The wording of this sentence is not quite right.

o   Page 6, line 203: It is not clear what “mostly positive effects” means. Also, reference which studies are being referred to.

o   Page 6, line 204: change “still” to “studies”. Again, it is not clear what “mostly no effects” means. Please clarify here or in the methods section.

o   In this section, it may also be interesting to know whether the effectiveness varied between the measures used – eg self-report vs objectively. Consider reporting this information as well.

o   Anthropometric measures by definition include all measures of the human body. In line 237 where “improvements in anthropometric measurements” is stated consider specifying what these were.

o   In line 239, “reduction in body fat” is noted. How was this measured?

o   In line 250, it is noted that “participants were already physically fit”. Were they physically fit or physically active?

o   There seems to be a word missing in line 252: “While not [all]…”

o   Were there any differences between school-based interventions that could have led to differences in effectiveness – eg were some multi-strategy vs single strategy; or how outcomes were measured. A comment about it may be useful here.

·         Table 2:

o   Given the focus on social ecological model, it may be useful to highlight in the table the levels each study targeted in their interventions as part of the “component” column.

o   Abbreviations have been included in the table. Make sure they are used consistently throughout. The is a mix of abbreviations and full written descriptions used (eg PA and physical activity).

o   The level of description of interventions in the “Components” column is not always consistent between studies.

o   Where possible, report how the outcome was measured with the outcome so it is clear. At times how they are measured are reported after the outcomes are listed.

o   Nyawose reference: sports is listed as an outcome. Has it been considered in reviewing effectiveness? If so, please state what about sport was measured eg participation in sports.

·         Table 3: The description in the footnote about how scores are given could be moved to the methods section. If left as a footnote to the table, consider making the description more succinct.

·         3.6. Implementation barriers and facilitators

o   Line 257: suggest adding “Barriers reported to implementing school setting interventions…”

o   Including some examples of overall barrier themes may be useful – eg for lack of resources, buy-in and motivation of whom?

o   Are the references at the end of the barriers sections relevant to all the barriers listed? It not, place relevant references against each individual barrier listed.

o   Line 262, sentence starting Walter: consider starting the sentence with “One study by Walter…” to make clear that Walter is one of the studies being discussed.

o   Line 269: end the final sentence “..with motivating teachers…” to be involved (or with what specifically is the issue).  

4. Discussion

·         Lines 300-303: When discussing the effectiveness of school-based interventions in this study, maybe do not state them being “the most effective”. You do not really have many interventions to compare them with. Maybe reword to being effective.

·         Paragraph 3, starting line 308: Is the purpose of this study about determining which behaviours to target? Epidemiological evidence clearly states that both diet and physical activity are important and both should be targeted in reducing obesity levels. How to intervene to influence each is what is important and possibly more the focus of this paper. Suggest reworking this paragraph.

·         Line 357: the importance of process evaluation is noted. What specific areas of process evaluation will be important in progressing this area of research?

·         Line 358: behaviour change theories are introduced for the first time. There is a wealth of evidence that supports the use of behaviour change theories to underpin the design of an intervention. However, this has not been assessed in this study. Either remove it, or introduce the role of behaviour change theory in the Introduction and assess it as part of the review process.

Author Response

Thank you for these helpful comments. Our detailed response is in the attached cover letter.

Round  2

Reviewer 3 Report

The authors have addressed my concerns and have clarified several points throughout. Thank you.